# Nurses’ Best Friend? The Lived Experiences of Nurses Who Utilized Dog Therapy in the Workplace

**DOI:** 10.3390/nursrep15070246

**Published:** 2025-07-03

**Authors:** Valerie A. Esposito Kubanick, Joy Z. Scharfman

**Affiliations:** 1Nursing Department, York College, City University of New York, Jamaica, NY 11451, USA; 2College of Nursing and Public Health, Adelphi University, Garden City, NY 11530, USA

**Keywords:** nurses’ burnout/stress, workplace anxiety, therapy/emotional support dogs, refueling/re-energized, positive impact on patient care, nurses’ relief, qualitative hermeneutic

## Abstract

The aim of this work is to explore and understand the lived experience of nurses who chose to schedule visits with an Emotional Support Animal (ESA), i.e., a dog, during their working shift. **Background/Objectives:** Nursing practice is rigorous, weighted with intense responsibility that creates an environment conducive to stress and anxiety for the nurses, who have reported a higher level of work stress than other healthcare professionals. Knowing and addressing the factors impacting mental health/nurses’ well-being is crucial to providing care to patients. Stressful work environments result in burnout, compassion fatigue, depression, anxiety, suicide, and resignation. Understanding nurses’ perspectives on work related stress encourages hospitals to structure practices supporting nurses’ mental health and ability to provide quality care. **Methods:** A qualitative hermeneutic phenomenological approach was employed. Participants scheduled six weekly 10-min visits with Rex, a certified Service Dog for anxiety depression and PTSD, and registered ESA. A sample of 11 RNs participated in Rex visits at the workplace; field notes were taken during observations of visits. Semi-structured 30-min interviews conducted via Zoom, audio-recorded, and transcribed with NVivo were conducted. **Results:** Data were examined with Interpretative Phenomenological Analysis. Four main themes emerged: Preparing for the unknown, Doing the work, Refueling and resetting-Visits with Rex, What about nurses? **Conclusions:** This study highlights the need for nurses and hospital systems to incorporate self-care/self-reflection, including time/opportunities for nurses’ stress management during their practice. Knowledge of nurses receiving ESA interventions sheds light on how to protect/preserve the well-being of nurses practicing in this demanding profession. ESAs for nurses in the workplace offers an option to utilize dog therapy to re-energize and continue their shift renewed and refreshed.

## 1. Introduction: Background of the Problem: Addressing and Mitigating Stress in Nursing

Professional nurses have reported a higher level of work stress than other healthcare professionals (Roberts & Grubb, 2014) [1]. Nursing practice is rigorous, weighted with intense responsibility that creates an environment conducive to stress and anxiety for nurses. Healthcare outcomes and the safety of patients deteriorate when nurses are exposed to situations that result in high stress and burnout. Therefore, restoring nurses’ harmony is imperative, and a creative solution is using canines for support. Animals have accompanied humans over many centuries. Dogs represent humanity’s oldest friend, offer unconditional love, and may be considered the perfect partner. The social bond between man and dog is extremely strong (Grimm, 2015) [2]. Florence Nightingale (1860) [3], the founder of modern nursing, remarked on the comforting power of animals with long-term patients. Nurses enter the profession with a desire to help people heal. Nurses are responsible for a large amount of patient care. Stress is intrinsic in caring for people struggling with health problems. Chronic job stress due to patient acuity, inadequate staffing, lack of time for self-care, and the depersonalization of healthcare organizations impacts nurses’ altruistic ability to provide quality care. Exploring creative stress-reducing, core-strengthening options for nurses is essential to preserve quality patient care and quality of life for professional nurses. Emotional Support Animals (ESAs) can be useful in all types of patient care and have been utilized in a variety of settings, including behavioral health, cardiac care, palliative care, hospice, rehabilitation/stroke, and pediatrics. ESAs provide their owners, or other individuals, with therapeutic benefits through companionship and can be any animal (Kubanick, 2023) [4]. The therapeutic benefit is derived from the companionship between the animal and the person. ESAs undergo extensive training to provide emotional support and comfort. ESAs frequently interact with other people besides their owner and are used in many different patient care areas therapeutically by healthcare providers to augment or enhance treatment plans (Kubanick, 2023) [4]. This study will explore nurses’ experience of work-related stress during work shifts and the impact of using an Emotional Service Animal, a dog named Rex, during break times. Gaining access to the nurses’ perspective can help hospitals to structure practices that support providers.

## 2. Literature Review

### 2.1. Stress/Coping

The nursing shortage has been a problem for decades. Lack of support, understaffing, and burnout contribute to this problem. Not having enough nurse educators fuels the continuation of this issue. Offering nurses meaningful support and comfort is essential. Connections with family, friends, and other positive relationships have helped nurses endure the pressure of their workload (Watson, 2024) [5]. A multitude of factors, individual, circumstantial, and systemic, impact nurses practicing in healthcare today. The shift in focus in contemporary healthcare settings to a business model that is fast-paced and task-oriented has affected nurses’ time and ability to interact with patients (Boeck, 2014; Scharfman, 2024) [6,7]. Devoting time to form a therapeutic relationship with a patient is often neglected (Heath, 2017) [8]. Regardless of the cause, the work environment is stressful and often results in burnout, compassion fatigue, depression, anxiety, suicide, and resignation (Melnyk, 2023) [9].

Stress is a complex phenomenon with multiple definitions. Stress is perceived as “a relationship between the person and the environment that is appraised by the person as taxing or exceeding his or her resources and as endangering wellbeing” (Folkman et al., 1986) [10]. The experience of stress is influenced by both cognitive assessments and coping abilities. Chronic stress may impair a nurses’ ability to function. Building resilience through self-care, positive thinking, and positive relationships can serve as a protective factor, enabling nurses to endure the high-pressure nature of their professional responsibilities (Tully, 2019; Watson, 2024) [5,11]. 

Coping capacity and strategies allude to both cognitive and behavioral attempts to master or tolerate a problem (Folkman & Lazarus, 1985) [12]. 

Coping tactics are designated as either problem- or emotion-focused. Problem-focused approaches aim to deal with the source of the stress, usually an external stressor. Emotion-focused coping is directed at modifying internal responses to stressors. Problem-focused coping involves changing the work environment while emotion-focused coping aims at making personal internal changes. A combination of problem- and emotion-focused coping was identified as optimal in the seminal study by Folkman et al. (1986) [10]. 

Both problem- and emotion-focused coping efforts are needed to ensure patient and nurse well-being. Increased demands of workload, efficiency, and focusing on tasks impacts nurses’ ability to dedicate time to the person-centered approach. Self-care, an emotion-focused coping technique, is crucial to preserve nurses’ ability to care for others. Adequate rest, breaks, nutrition, hydration, and space to decompress are essential to refuel nurses during an 8- or 12-h shift. Inadequate opportunities for stress reduction can challenge a nurse’s capacity to be present with patients and lead to decreased engagement and increased risk for errors. 

The challenge to provide safe, quality care with highly acute patients and insufficient staffing takes a toll on the motivated professional nurse (Akbar et al., 2017) [13]. 

### 2.2. Emotional Support Animals in Healthcare

SDs are trained for specific tasks for their handler’s (owner’s) needs. These highly trained dogs supply the function/replenishment that the individual needs to be able to have the fullest life possible. SDs may be trained to perform many tasks depending on the owners’ disabilities. Some specific tasks include guiding individuals who are blind; alerting deaf individuals; assisting people with diabetes, seizures, or an unsteady gait; and tasks for people who are para- or quadriplegic. SDs help meet needs to strengthen quality of life (Lundqvist et al., 2019) [14]. SDs are registered by state but are classified differently than Emotional Support Animals (ESA).

A systematic review of 20 studies by Lundquist et al., (2019) [14] showed that at least 15 of the inquiries revealed a significant positive effect of the use of dog-assisted therapy when treating adults and youth in a psychiatric setting. According to Rodriguez et al. (2018) [15], the therapeutic value of using a service dog has been reported. Primarily, SDs were used to help treat anxiety, depression, and Post-Traumatic Stress Disorder (PTSD), particularly in military personnel (Kubanick, 2023) [4]. A study by Jensen et al. (2020) [16] examined the use of a facility dog for healthcare professionals in a pediatric setting. Using two groups of participants (N = 130), the researchers had one group (n = 65) spend time with canine assistants while the other group (n = 65) did not. Data was collected using the Malbach Burnout Inventory: subscale for emotional exhaustion, the Job-Related Depression Enthusiasm Scale, the Job in General Scale, the Workplace Social Support Scale, Anticipated Turnover Scale, Turnover Intention Scale, the Patient-Reported Outcomes Measurement Information System scale, and the Scale of Positive and Negative Experience. Results showed evidence of stress relief, greater job-related enthusiasm (*d =* 0.48) for the healthcare providers who spent time with dogs, and better perceptions of the job overall with a large effect size (*d* = 0.57). Similarly, Gerson et al. (2023) [17] examined burn out and resiliency in clinical and non-clinical staff in a pediatric doctor’s office using dogs to visit the office for one session per participant. Quantitative analysis showed that nearly half of the 149 participants scored higher on readiness to work after having a dog session (M = 78.1, SD = 18.4). Using the State Anxiety Scale Mulvaney-Roth et al., (2022) [18] were able to examine anxiety in various hospitalized patients with the use of therapy dogs. Through experimental quantitative design (N = 54) the results showed a positive effect on lower anxiety in both adults (n = 42) and pediatric patients (n = 12) after the use of therapy dogs (M = 2.73, *p* = 0.002), with an effect size of 0.64.

Research conducted in Japan by Murata-Kobayashi et al. (2023) [19] explored the use of nurse handlers and hospital dogs accompanying the nurses when they tended to pediatric patients. Other healthcare disciplines (N = 626) have also used hospital dogs, with researcher-developed surveys were completed after patient visits. The researchers found that the highest rated items were “impact on terminal care” and “patient cooperation” after visits from a professional when accompanied by the hospital dog. A study by Townsend et al., (2022) [20] analyzed dog visits to inpatient adults, children, adult visitors, and healthcare workers in an acute care setting during COVID-19. Visits from the dog varied with each interaction including observations (N = 1016), and most visits were with healthcare workers; however, patients had longer visit times (M =5.81, SD + 4.38). The researchers concluded that having the dog available was an effective way to reach large groups of people across the hospital setting. Therefore, dogs were able to safely visit during the pandemic. 

Machova et al., (2019) [21] studied the use of dogs present in the workplace and the possible effects on the level of stress in nurses. This was performed by measuring cortisol levels through saliva samples of the nurses (N = 22) during a regular work time, then during a break and thirdly during a break with a dog visit. The results showed a significant difference in the reduction in cortisol levels (n = 19), suggesting a decrease in stress. In a mixed methods study by Etingen et al. (2020) [22] an animal- assisted program was implemented in a Veteran’s hospital for multidisciplinary use (N =39). After twenty sessions conducted over a three-month period, employees reported lower levels of burnout and significantly improved moods using both quantitative—the Mood Scale Survey (M =18 vs. before, M =40)—and qualitative findings, suggesting that the employees were highly satisfied with the animal-assisted program. Steinberg et al. (2024) [23] used therapy dogs to examine the impact of animal assisted activity on healthcare worker stress, burnout, work engagement, and mood in a quasi-experimental study. The program was designed to increase access to the therapy dogs and visits did not require a scheduled time commitment, offering the availability of a supportive connection based on the individual’s workload and staffing. The authors report a significant improvement in the mood of the healthcare workers and suggest that improved mood may translate into improved patient care and satisfaction (Abrahamson et al., 2016; Kang et al., 2019) [24,25].

Therapy dogs, whether ESA or SD, are used by healthcare providers to augment treatment plans (Levey et al., 2017) [26]. Dogs have a unique and powerful bond that can harness relationships as a treatment modality in behavioral health. Routine visits to inpatients and outpatients by ESAs are also well received. Therapeutic communication flows effortlessly between healthcare providers and the patient when the ESA is at the bedside. Patients anticipate the ESA’s visits, and the dog is the social catalyst that evokes immediate smiles and relaxation upon initial contact. Animals have been in a progressive role as used in the treatment of a diverse number of psychiatric disorders (Fine, 2010) [27]. The benefit of healthcare providers collaborating with handlers is for the betterment of the patients. Animal-assisted therapy with children, adults, and elderly patients during counseling sessions and psychotherapy has been successful in treating a variety of disorders such as anxiety, depression, or social impairment. Specifically noted in the treatment of autism patients was the use of SDs as a vehicle for socialization. A study by Winkle et al., (2019) [28] that examined the use of service dogs as a treatment for adolescent patients with developmental disabilities and their parents. The researchers found that the use of service dogs helped decrease the patients’ anxiety during typical daily functioning and increased the patients’ social interactions outside of the home. 

### 2.3. Significance of the Study 

This study aims to explore the lived experience of nurses who deliberately chose to schedule visits with an ESA during their working shift. Research exploring the use of time spent with a service dog on the well-being of nurses is scant (Watson, 2024) [5]. Animal-assisted activities have traditionally focused on the effect on the patient, leaving the nurse out of the picture and revealing a gap in the literature. A recent systematic review of the literature that examines the benefits of Animal-Assisted Interventions (AAIs) on healthcare workers revealed research exists (Acquadro Maran et al., 2022) [29]. Nurses, who provide vital care and a unique role in the healthcare team, were included in the group, but were not the focus. Understanding the experience of nurses receiving Animal-Assisted Interventions may shed light on how to protect and preserve the well-being of individuals practicing in this unique, demanding profession. This study aims to understand the impact of six weekly scheduled visits with a therapy dog in a room off the unit. The nurses scheduled visits with the researchers and the service dog during a shift they were working. 

## 3. Materials and Methods

This is a qualitative hermeneutic approach to examine the lived experiences of nurses that utilize an ESA during work shifts in an acute care setting, a private inpatient hospital. Phenomenological inquiries aim to explore the internal experience of nurses. Interpretive phenomenology is founded on the concept that examination of the experience guides one to core knowledge. This method is associated with Heidegger’s [30] approach, which highlights how this examination leads to questions about the world and the objects within it. Understanding the impact of planned visits with Rex, a Certified Service Dog and Emotional Support Animal, during their practice may reveal how this opportunity may influence nurses’ capacity to sustain their energy, motivation, and focus on patient care during their assigned shift. Research aimed at preserving nurses’ sense of wellness during practice is essential to preserve and retain members of the profession. 

This study utilized a dog, Rex, a certified Service Dog for anxiety depression and PTSD and a registered Emotional Support Animal. However, Rex’ role was an Emotional Support Animal for the nurses to explore stress reduction. Flyers were posted throughout the hospital for the recruitment of volunteer participants. Nurses with canine allergies were excluded. A purposive sample of 11 registered nurses currently working were included, undergoing visits and interviews. Six 10-min weekly visits with Rex were scheduled by the nurses and researchers according to their work schedule. The nurse participants visited one-on-one with Rex in a staff lounge. The researchers took field notes during observation of the nurses interacting with Rex. Rex was eager to “work” with the nurses in stress reduction. He was calm, happy, and excited to see the nurses for their individual visits. No more than two visits per day were scheduled. Semi-structured interviews were conducted by both researchers with each participant on Zoom until data saturation was obtained at interview seven. This was noted as no new information was obtained upon the last interview. Demographic data was obtained at the time of each interview (see Table 1). Each interview, lasting approximately 30 mins, was audio-recorded and transcribed. Open-ended questions and probes were used, allowing the researchers to follow the participants’ narrative and ask for clarification to acquire a comprehensive description of each nurse’s lived experience (Streubert & Carpenter, 2011) [31]. Nurses were asked to describe their feelings before, during, and after working a shift, their experience of stress, and the impact of the visits with Rex on them. The question of whether regular visits from a support animal would be desired and/or beneficial was posed (see Appendix B).

COREQ (consolidated criteria for reporting qualitative research) Checklist has been completed, see Appendix A.

### 3.1. Protection of Human Subjects

Before commencing this research, approval was obtained from both the Institutional Review Board (IRB) of the City University of New York and Catholic Health. Verbal and written informed consent, confidentiality, freedom to withdraw, risks, and benefits were discussed before data collection. Permission to audio-record the interview was obtained. Audio recordings and transcripts were coded and secured for privacy and confidentiality. Nurse participants were given fictitious names (Nurse A-K). Participants were informed of the possibility that the study may be published; however, no identifying information would be used. The researchers remained sensitive and attuned to any signs of emotional distress during the interview and safeguarded against undue stress, emphasizing the assurance of confidentiality and the voluntary nature of participating in the study.

The participants offered treats and water, provided by the researchers, to Rex during some of the visits, which he happily took. Rex was certified, vaccinated, and in good health and spirits before, during, and after the study.

### 3.2. Data Analysis

Demographic data obtained was used to describe the participants in the study. The audio recordings and verbatim transcripts were used as data to be analyzed. NVivo software (NVivo 15, 15.01.1) was used for audio transcription and coding. Interpretative Phenomenological Analysis (IPA), as outlined by Smith et al. (2022) [32], was the method chosen for data analysis. This method was chosen as it aligns with the philosophical underpinnings of hermeneutic phenomenology. A cyclical process of self-reflection and self-reflexive journaling was initiated prior to data collection and continued throughout the research, focused on knowing personal ideas and attitudes and developing an awareness of and dwelling on pre-understandings and fore-projections/fore-structures. Field notes of each visit were recorded. Immersing oneself in the original data is the first step of an IPA. Active engagement with the data begins with reading and rereading the first interview transcript and listening to the audio recording to develop a sense of familiarity with the phenomenon, enabling the researcher to better recognize and take note of the common threads of understanding. Line-by-line reading of the transcript and underlining of keywords and phrases led to writing exploratory notes in the margin of the transcript related to the highlighted segments. The next stage entails consolidating, analyzing, and extracting the most important features of exploratory notes into experiential statements. These statements, written in the opposite margin, reflect a description and interpretation of crucial points captured from the text. The process continues with developing a map or chart of how the statements fit together to form clusters. Clusters are organized into a table where they are named Personal Experiential Themes (PETs). A Word document created by the researchers was reviewed until Group Experiential Themes (GETs) emerged as possible themes of the narratives. The PETs were organized into the sub-themes of the GETs. The process was dynamic and involved moving material around and searching for strong, meaningful connections.

**Table 1 nursrep-15-00246-t001:** Demographic data of nurse participants.

Categories	Levels	Totals
Age	20–29	6
	30–39	5
	40–49	
	50–59	
	60 or more	
Gender	Male	1
	Female	10
Ethnicity/Race	Indian	4
	Caucasian	5
	Black	1
	Asian	1
Degrees	Associate’s degree in nursing	3
	Bachelor’s degree in nursing	8
	Master’s degree in nursing	
Years as an RN	1–2	4
	3–5	5
	6–10	2
Clinical specialties of practice	Emergency Room	7
	Medicine	1
	Surgery	1
	Intensive Care Unit	2

### 3.3. Methodological Rigor

The methodological rigor of this study is achieved by adherence to the criteria to develop trustworthiness in qualitative research: credibility, transferability, dependability, and confirmability proposed by Lincoln and Guba (1985) [33]. Credibility was determined by prolonged engagement with the data, multiple readings and listening to the interviews, and researcher triangulation, reviewing each interview and emerging themes with an advisor. An audit trail demonstrating an example of the movement from verbatim phrases to themes was produced to establish the dependability of research findings. Detailed descriptions of the data may enable transferability. Dependability was achieved by a thorough line-by-line review of each transcript according to the 7 steps of the Interpretative Phenomenological Analysis (IPA) method outlined by Smith et al. (2022) [32]. Reflexive journals and field notes were kept by the researchers throughout the process of inquiry.

## 4. Results

Both researchers were involved in the coding, and inter-rater reliability was determined. The participants concurred that stress was an inherent part of nursing. Many participants shared about anxiety prior to working, worrying about adequate staffing, patient load and acuity, or being floated to another unit. During a shift, the same factors influence the stress level of the nurse, and visits with Rex allowed them to take a few minutes off, relax, refocus, regroup, and return to caring for their patients with renewed energy and improved mood. All the nurses reported providing better care for their patients after a visit with Rex. Four themes emerged from the data: Preparing for the unknown, Doing the work, Refueling and resetting, and What about nurses?

### 4.1. Theme 1: Preparing for the Unknown

Preparing for the unknown refers to a variety of factors nurses think about prior to their shift. Each of the participants referred to pre-shift anxiety related to concerns about adequate staffing, unwieldy patient load, high patient acuity, sick calls, codes and deaths, and being floated. Several nurses remarked: “Every shift, I have no idea what I’m walking into.” Participants: Nurse I, Nurse C, Nurse K. One nurse practicing in the emergency room (ER) explained:

“It’s my pre-shifting anxiety stress. You worry if you’re going to be short staffed, if people call in sick, if there’s going to be a very high census of patients. Then when you are working, it can be stressful if you have a lot of patients and not a lot of time to do anything but patient care. You can’t drink water. You can’t go to the bathroom. You can’t even sit down. And it’s really depending on how sick the patients are, if you have really sick patients, that can be stressful. Then when you go home, if you had a patient that didn’t make it, or if you had a really tough case, that sticks with you. You think about these things when you go home, too.” (Nurse I)

There is ample literature describing work-related stress and nurse burnout. Unchecked burnout has driven many nurses to leave the profession. High stress levels impact the quality and safety of care (Brox, 2019; Gerson et al., 2023; Jensen et al., 2020) [16,17,34]. Multiple factors contribute to the stress experienced by a nurse, who, by virtue of the profession, has significant responsibility for human lives. Staffing levels, adequate resources, support, and the climate of the institution influence nurses from a systems level. The acuity of patients and the patient census fluctuate and demand interdisciplinary collaboration and clear lines of communication. One nurse shared the following:

“Work takes a lot of me. I start preparation for work the night before. I have a winddown routine, a time that I aim to fall asleep by. Just so I make sure that I get 7, 8 h of sleep.” (Nurse B)

It was evident that the nurses who participated in this study realized that their role demanded focus and attention which they prepared for both physically and emotionally. Anticipatory pre-shift stress was an experience shared by every participant, reflecting the gravity of the scope of the professional practice of nursing.

### 4.2. Theme 2: Doing the Work

Doing the work encompasses what actually occurs during a shift. Several participants practiced in the emergency room, which they described as “a very high stress environment”. The flow of patients and the care each patient requires may vary from manageable to hectic. One ER nurse related the following:

“There are times that work gets so stressful. You just keep Doing one thing after another after another and you get in the mindset of it and you just feel very stressed out.” (Nurse I)

Another ER nurse shared the following:

“Working in the ER is a very high stress environment. You must easily be able to adapt to changing. Oftentimes, you can have multiple patients and the stress is building. One specific time, we had a code, a neuro happening. Just getting through the whole code neuro protocol, my stress was still very high at the time.” (Nurse H)

One nurse working on an oncology unit reported the following:

“Well, every shift, I don’t know what I’m walking into. Especially sometimes I’m over contracted overnight, so it could be stressful, having more than six patients. It could be good and then it could get crazy. It really depends.” (Nurse C)

Another nurse practicing on a medical-surgical unit with a respiratory focus/ventilator capability described stress from things that interfered with patient care:

“There are constant interruptions, from the telephones of family members, other staff members asking questions. So, it’s a lot to balance. Balancing that with, of course, attending to the patients. Distractions that take your focus away from the reason you’re there- to take care of the patients.” (Nurse B)

The reality is that there is no way to predict or control what happens during a shift. Knowing that nurses are often faced with handling very stressful situations, as revealed by the narratives of the participants in this study, highlights the need for stress relief. Offering multiple opportunities for nurses to regain their energy and focus is imperative. Ease of access and flexibility are essential for nurses to utilize “de-stress” options. 

### 4.3. Theme 3: Refueling and Resetting-Visits with Rex

Each participant shared that visits with Rex were beneficial in many ways. First, connecting with Rex meant they had a break. Taking a break was not easy during a stressful shift, so the planned visits prompted nurses to deliberately take time to step away. One participant commented on the following:

“It was a nice break from all that. At times it was a forced break which you need sometimes. You had to put things down and go see Rex. It was enjoyable. It was a little bit of a mind reset. Let go of all that stuff for 10 mins or so. Connecting with the dog, force yourself to take a break from the stress. Unplug from everything coming at you, then go back, it helps prioritize better, get a new start.” (Nurse B)

Acquadro Maran et al. (2022) [29] reviewed the literature on animal-assisted intervention and healthcare workers’ psychological health, pointing to many studies in which time spent with a pet could serve as a “stress buster”, decreasing anxiety and hyperactivity, increasing self-esteem and self-efficacy, and encouraging feeling positive emotions, all of which enhances coping strategies for stress management. Another nurse stated the following:

“Being able to step away for a few minutes was actually helpful, especially in those high stress and difficult situations. He made it a little more bearable. I was able to regroup and go back to the patients and continue care for the day, After the visits, I felt calm, more relaxed, more focused. I feel like it helped me give better care.” (Nurse A)

Many participants shared that visiting with Rex improved their mood. One ER nurse shared the following:

“Being able to step away from all that and recoup always made me so much happier. And then being able to go back, I just felt more focused and less stressed and able to provide better care because of that.” (Nurse I)

Connections and supportive relationships were identified by nurses as protection against compassion fatigue and promoters of good professional quality of life (Watson, 2024) [5]. Each participant spoke of their special feelings for Rex during the visits, with several stating “I’m so happy! I love you!”, “I definitely need you today”, “I can feel the tension running out of my body”, “You make me feel so calm”. The nurses referred to Rex as “such a beautiful, gentle good boy”. The participants remarked “this is exactly what we needed today”, on multiple visits. Barker et al. (2005) [35] discuss observations of positive attitudes and smiles in healthcare providers who were able to take a break from a hectic workplace, suggesting that positive emotions can be associated with health-promoting effects. Several studies report on improved moods, decreased burnout, and greater work-related enthusiasm among healthcare workers participating in programs using therapy dogs (Etigen et al., 2020; Jensen et al., 2021) [16,22]. One ER nurse shared the following:

“Being with Rex helped improve my mood, lowered my stress levels. He even reduced my loneliness, because sometimes working with patients and the patient load, you get so overwhelmed, even though you’re surrounded by people. It can also be lonely. You’re the nurse responsible for the total care and collaboration of this patient. It can get so lonely. And he increased my motivation to get back to work with an increased activity level, too.” (Nurse H)

Every participant spoke of the positive impact that visits with Rex had on their patient care.

### 4.4. Theme 4: What About Nurses?

The focus of research on the impact of pet therapy is on the well-being of the patient. Clearly, patients are a primary concern when exploring healthcare. Positive patient outcomes and patient satisfaction are the priorities of professionals practicing in the healthcare setting. The literature now includes research implicating the stress and burnout of healthcare workers, particularly nurses, as significant in positive quality and safe patient care. All the nurses in this study spoke of this issue. One nurse remarked the following:

“Since I’ve been in health care, the main focus for emotional support animals has always been with the patients. So, I think it’s really great to do it for the health care providers, too. I think it would be really great if we were able to consistently utilize that in the hospitals, for health care providers, and patients, for everyone.” (Nurse G)

This theme supports the need to offer both problem- and emotion-focused coping for nurses. Scheduling a visit with Rex involves both external and internal strategies. The visit was perceived as a “forced break” by the nurses, a system or environment modification. The change in mood and stress-relief represents an internal or emotion-focused coping action. Once able to detach from the work stress, spending ten minutes with an emotional support dog was experienced as therapeutic. When asked if there would be any benefit from regular visits from an ESA, one of the ER nurses asserted the following:

“I think it would be wonderful. They have an aromatherapy setup. It’s a quiet, low light, calming environment. Having a dog like Rex on a regular basis, I would enjoy that, too, even just an unscheduled visit, because working in an unpredictable environment like we do, it can’t always be scheduled. So, you might find yourself with a free five or ten minutes and if you could go and get your “Rex fix”, so to speak. It would be very nice.” (Nurse H)

This sentiment supports the importance of the connections, which are deemed vital positive coping behaviors, to improve nurses’ professional quality of life (Watson, 2024) [5]. Preserving nurses’ well-being is critical for safe, quality patient care and satisfaction for both patients and nurses. Nurse satisfaction plays a pivotal role in reducing burnout and the volume of nurses choosing to leave the profession (Gerson et al., 2023) [17]. Figure 1 demonstrates the Nurse/Rex relationship and themes.

## 5. Discussion

The purpose of this study was to explore the experiences of nurses in the workplace who had the opportunity to have regularly scheduled visits from an Emotional Support Animal, Rex the dog. The findings confirm that regular visits from Rex for nurses in the workplace were truly valued. Benefits for nurses who experienced the therapeutic visits revealed a sense of decreased stress, lowered anxiety, refueling of energy and purpose, and motivation to continue high-quality patient care for the nurses. The four main themes emerged from the data to describe the nurses’ experiences of connecting to Rex, a supportive, loving companion. Support for these findings is evidenced in similar nursing literature, and research. The themes identified through this research may aid in understanding nurses’ perspectives regarding stress while working in an acute care setting. Additionally, nurses’ self-appraisal of stress in congruency with utilizing the dog therapy may assist the nurses in identifying coping strategies to enhance their practice and improve patient care.

Nurses in the workplace greatly benefitted from therapy dog breaks. The narrative of the participants illustrated in the themes is supported by the theory of Lazarus and Folkman (1984) [36], with states has a model that depicts the relationships between an individual’s (nurses) recognized psychological stress, coping, and cognitive perception and understanding. All the nurse participants shared a high level of stress that was mitigated with regular visits from Rex. This study revealed a humanistic element; the ability of the nurses to be cared for and then to continue caring for their patients. This relates to the Humanistic Nursing Theory (Paterson & Zderad, 1976) [37], a theory that believes that patients can grow in a healthy and creative way. The findings of this study on Rex comforting the nurses, also aligns with Benner’s theory (1984) [38] which asserts that being with a patient is itself therapeutic. The comfort that Rex provided gave strength and renewed purpose for the nurses to care for their patients.

Comparable studies with nurses and dogs in the workplace are very limited. A systematic review of the literature by Aquandro Maran et al. (2022) [29] found that using animal assistance for healthcare workers to reduce stress was beneficial and desired by all healthcare workers. A study by Steinberg (2024) [23] used animal-assisted activities for healthcare workers that included animal visits during work. This was found to be beneficial and helped to decrease stress and burnout. Barker et al. (2005) [35] conducted a quantitative study that measured cortisol levels in saliva and blood serum before and after nurses visited with therapy dogs during their work shift. Salivary and serum levels of cortisol were significantly lower after 5 and 20 min interactions with a therapy dog. This was a biological indication that the use of therapy dogs decreased levels of stress for the nurses.

### Limitations of the Study

A limitation of interpretive research is that it reflects the interpretations of the researchers and the study participants, which may not provide a comprehensive representation of the population. Additionally, a convenience sample of nurses from a single setting institution was utilized. Conducting interviews on Zoom rather than in person may have influenced the level of sharing during the interview process. Nurses allergic to dogs were ineligible to participate.

## 6. Conclusions

This study highlights the need for nurses and hospital systems to incorporate self-evaluation, self-care, and time for re-energizing into their practice. Regular dog therapy programs, such as having dogs present in the hospital for various shifts, should be part of hospital administration plans to assist nurses while working in stressful environments. Nurse leaders can influence practice environments and support nurses in having outlets for workplace stress. This will allow the nurses to practice patient care safely and reduce burnout. Having the availability of an ESA, such as a dog, for nurses in the workplace would offer an opportunity for nurses to deliberately utilize self-assessment and self-care into their workday. There is data available that points to positive outcomes with dogs in healthcare service. The need for nurses’ mental health and stamina exists, and dogs are willing to love, serve, and help improve nurses’ overall well-being (Kubanick, 2023) [4]. An unknown author once said *To err is human, to forgive is canine.*

### 6.1. Implications for Nursing Education

The importance of self-assessment, self-care, and preserving the ability to provide quality care despite stressors needs to be included in nursing education. Additionally, education regarding the therapeutic benefit of dogs in the workplace for nurses is essential for pre-licensed and licensed nurses. 

### 6.2. Implications for Practice and Administration

Our study seeks to help nurses practice patient care safely and reduce burnout. Administrators can explore and evaluate obstacles such as inadequate staffing and resources, which may impede the nurses’ ability to provide safe quality care for patients. Our study provides a medium to facilitate the nurses’ well-being in high stress situations.

### 6.3. Implications for Research

Research focusing on how nurses cope with job stress is important to preserve the health of the profession and the world. Nursing is a demanding profession that requires specific skills, alertness, concentration, time management, critical thinking, and judgment while caring for patients in any phase of living and dying. More research about regularly planned dog therapy for nurses in the workplace should be explored in a variety of healthcare settings. 

### 6.4. Relevance to Clinical Practice

This study highlights the need for nurses to have an opportunity to destress during work hours. Knowledge of nurses receiving ESA (dog) therapy in the workplace may shed light on how to protect and preserve the well-being of nurses practicing in this idiosyncratic, taxing, and depleting profession. Having an ESA available for nurses in the workplace could offer an option for nurses to utilize dog therapy to re-energize and continue their shift renewed and refreshed. Nurses, with workplace self-care, would thereby pass along this benefit when providing care to their patients.

## Figures and Tables

**Figure 1 nursrep-15-00246-f001:**
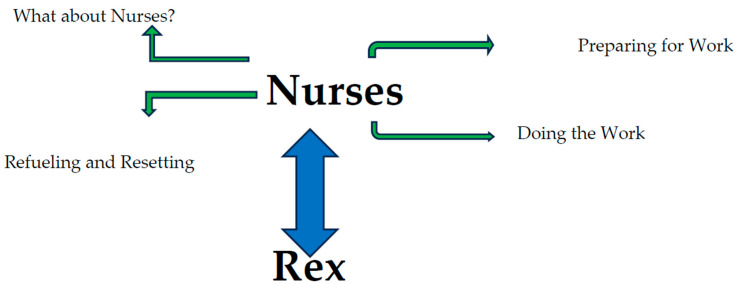
Nurse/Rex Model.

## Data Availability

Data not available due to privacy restrictions.

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
