# Peer review of "Nurses’ Best Friend? The Lived Experiences of Nurses Who Utilized Dog Therapy in the Workplace"

_nursrep, 2025, doi:10.3390/nursrep15070246_

Round 1
Reviewer 1 Report
Comments and Suggestions for Authors
Dear authors,
I find this article interesting because any activity that reduces the stress inherent in the work of healthcare professionals is necessary to ensure adequate healthcare.
The title provides adequate information about the topic covered and would facilitate searching databases.
The abstract provides a comprehensive, detailed, and clear overview of the study, although I believe it should indicate the study's objective and sample size.
The introduction broadly addresses current issues regarding the research topic with extensive bibliographic references.
The objective of the study is clearly defined.
The methodology is sufficiently detailed to allow for reproducibility of the research. However, I believe the questions asked of participants should be more clearly indicated, including a table.
It is not clear to me whether the participants were visiting the patients accompanied by the dog or whether they spent time with the dog between patients. This aspect should be clarified.
The sample is quite small, only 8 participants, so the possible consequences of this should be indicated in the limitations section.
Why wasn't some specific program for handling qualitative data, such as MAXQDA, used for the analysis?
I think Table 1 should be included in the results section.
Table 1 only details the demographic characteristics of 7 participants, when previously stated there were 8. What is the exact number of participants?
In the results section, I think more participants' responses should be included, since there are so few participants, so that the reader can have a more precise view of the answers. Only one participant's response is indicated. Did they all answer exactly the same thing?
The discussion seems limited to me, and I think it should be expanded to compare the results obtained in this study with the existing evidence on the research topic.
The limitations should include those derived from the type of sampling used, the small sample size, and the possibility of conducting a quantitative study to assess stress before and after the intervention.
The conclusions are consistent with the results obtained and meet the proposed objective.
The references are adequate and up-to-date.
Kind regards.
Author Response
Comment 1:The abstract provides a comprehensive, detailed, and clear overview of the study, although I believe it should indicate the study's objective and sample size.
Response: nurses’ perspective on work related stress encourages hospitals to structure practices supporting nurses’ mental health and ability to provide quality care, this is explained under the heading of objective in the abstract keeping in mind we had to adhere to a word count. sample size in abstract Sample of 11 RNs participated in Rex visits at the workplace;
Comment 2: The methodology is sufficiently detailed to allow for reproducibility of the research. However, I believe the questions asked of participants should be more clearly indicated, including a table.
Response 2: Attached is a supplemental file with interview questions
Comment 3: It is not clear to me whether the participants were visiting the patients accompanied by the dog or whether they spent time with the dog between patients. This aspect should be clarified.
Response: Sample of 11 RNs participated in Rex visits at the workplace; field notes were taken during observations of visits, from abstract. The nurses were working a shift and met with Rex privately for 10 minutes. The nurse participants visited one-on-one with Rex in a staff lounge
Comment 4: The sample is quite small, only 8 participants, so the possible consequences of this should be indicated in the limitations section.
Response: 11 participants for visits and 8 were available for one - on -one interviews after completion of all visits, however field notes were taken for all participants. Participants were interviewed until saturation as per interpretive phenomenological data analysis.
Comment 5: Why wasn't some specific program for handling qualitative data, such as MAXQDA, used for the analysis?
Response: NVivo software was used for audio transcription and coding. Interpretative Phenomenological Analysis (IPA), as outlined by Smith et al.(2022); stated in data analysis.
comment 6: Table 1 only details the demographic characteristics of 7 participants, when previously stated there were 8. What is the exact number of participants?
Response: table adjusted to include all 11, although 8 were interviewed
Comment 7: In the results section, I think more participants' responses should be included, since there are so few participants, so that the reader can have a more precise view of the answers. Only one participant's response is indicated. Did they all answer exactly the same thing?
Response: results include 6 different nurses' responses, coded by letters A through I
comment 8: The discussion seems limited to me, and I think it should be expanded to compare the results obtained in this study with the existing evidence on the research topic.
response: Very scant studies available that are comparable, studies from literature review included in discussion.
comment 9: The limitations should include those derived from the type of sampling used, the small sample size, and the possibility of conducting a quantitative study to assess stress before and after the intervention
Response: sample size indicated as convenience added to the limitations. Size adequate for saturation. This was a qualitative research study.

Reviewer 2 Report
Comments and Suggestions for Authors
The study explores the impact of therapy dog (Emotional Support Animal – ESA) visits on reducing work-related stress and enhancing professional motivation among nurses. This topic is highly relevant and timely, given the intense stress experienced in the nursing profession today.
Below are my suggestions:
-
The introduction is quite lengthy and sometimes overly detailed. Particularly, the literature review section takes up too much space. It would be more effective to introduce the core problem of the study earlier. The transition to animal-assisted therapies should be presented earlier and directly connected to the nursing context. The effects of animals on humans are explained in excessive detail.
-
Some expressions are too colloquial (e.g., “Dogs represent humanity’s oldest friend...”, “offer unconditional love”), which is inconsistent with academic writing style.
-
The nurses’ stress, burnout, and reasons for leaving the profession are repeated across different paragraphs and could be simplified.
-
The referencing style throughout the manuscript is incorrect. For example: “and suggest improved mood may translate into improved patient care and satisfaction (Abrahamson et al., 2016; Kang et al., 2019) [23, 24]. Therapy dogs, whether ESA or SD, are used by healthcare providers to augment treatment plans (Levey et al., 2017) [25].” The references appear both numerically and by author/year, which is redundant and inconsistent. This should be fixed throughout the paper.
-
Towards the end of the introduction, there is an extensive description of the methods (e.g., "six weekly visits with Rex..."). This information should be moved to the Methods section, keeping the introduction focused only on the research aim and context.
-
The introduction should be shortened and simplified into 3–5 paragraphs. The literature review should be presented more systematically.
-
The ending of the introduction should be clearer and more impactful. The research gap, question, and contribution of this study should be summarized in a brief paragraph.
-
Why was a sample size of 8 considered sufficient? How was data saturation determined? This should be explained.
-
The details of the visits or environmental conditions were not described. Under what conditions did the visits take place? Were the nurses alone during the visits? What was the dog’s position? Such factors could influence interpretation. Was Rex’s behavior observed? Or only the nurses’ emotional states?
-
Examples of the interview guide and key questions should be provided.
-
There is no ethical consideration regarding the welfare/impact on the dog. Ethical considerations in such research should be dual-sided (both human and animal).
-
It is not stated whether multiple researchers were involved in coding. Was inter-rater reliability or independent checking done?
-
Writing errors should be reviewed, e.g., “form” → “from” (line 393); “aa” → “as” (line 401), etc.
-
The discussion could be enriched by including more recent studies or addressing conflicting literature if present.
-
The limitations section could be expanded, mentioning sample size, single-institution setting, or generalizability of findings.
-
Conclusions and recommendations could specify how often and under what conditions therapy dog visits would best serve nurses.
Author Response
-
The referencing style throughout the manuscript is incorrect. For example: “and suggest improved mood may translate into improved patient care and satisfaction (Abrahamson et al., 2016; Kang et al., 2019) [23, 24]. Therapy dogs, whether ESA or SD, are used by healthcare providers to augment treatment plans (Levey et al., 2017) [25].” The references appear both numerically and by author/year, which is redundant and inconsistent. This should be fixed throughout the paper.
Response: References were included as per APA 7th edition. However, the numbered references were requested by Nursing Reports and we added them in.
Comment 1- Introduction edited. Highlights explain animal-assisted therapy in the nursing context.
Comment 2- Citations provided.
Comment 3- Nursing stress information simplified and streamlined in literature review.
Comment 4- References APA 7th edition style. Numbers were added as per Reviewer 1's comments.
Comment 5- Information moved to methods section.
Comment 6- The Introduction is simplified, and the literature review is systematically organized.
Comment 7- Summary at the end of the introduction.
Comment 8- Sample explained on lines 185-192.
Comment 9- Explained in the materials and methods section. The dog's disposition is addressed in methods.
Comment 10- Addressed in methods, line 198.
Comment 11- Addressed on lines 210-212.
Comment 12- Addressed in results, line 261.
Comment 13- Edits completed
Comment 14- Discussion enriched with other studies- line 451-459 and throughout the discussion.
Comment 15- Limitation section revised.
Comment 16- Conclusions and recommendations revised starting line 470.
We are attaching the edited version with highlighted changes.

Round 2
Reviewer 1 Report
Comments and Suggestions for Authors
Dear Authors,
I consider that the manuscript has been sufficiently improved and I have no additional comments or suggestions.
Kind regards.